# Optimal Nozzle Position and Patient’s Posture to Enhance Drug Delivery into the Peritoneum during Rotational Intraperitoneal Pressurized Aerosol Chemotherapy in a Swine Model

**DOI:** 10.3390/jpm12111799

**Published:** 2022-10-31

**Authors:** Dong Won Hwang, Eun Ji Lee, Joo Yeon Chung, Eun Joo Lee, Dayoung Kim, Soo Hyun Oh, Seungmee Lee, Ga Won Yim, Seung-Hyuk Shim, Sung Jong Lee, San-Hui Lee, Ji Won Park, Suk-Joon Chang, Kyung Ah Pak, Soo Jin Park, Hee Seung Kim

**Affiliations:** 1Department of Obstetrics and Gynecology, Seoul National University Hospital, Seoul 03080, Korea; 2Department of Obstetrics and Gynecology, Chung-Ang University Hospital, Seoul 06973, Korea; 3Department of Obstetrics and Gynecology, Gil Medical Center, Gacheon University College of Medicine, Incheon 21565, Korea; 4Department of Obstetrics and Gynecology, Keimyung University School of Medicine, Daegu 42601, Korea; 5Department of Obstetrics and Gynecology, Dongguk University College of Medicine, Goyang 10326, Korea; 6Department of Obstetrics and Gynecology, Research Institute of Medical Science, Konkuk University School of Medicine, Seoul 05030, Korea; 7Department of Obstetrics and Gynecology, Seoul St. Mary’s Hospital, College of Medicine, The Catholic University of Korea, Seoul 06591, Korea; 8Department of Obstetrics and Gynecology, Wonju Severance Christian Hospital, Yonsei University College of Medicine, Wonju 26426, Korea; 9Department of Surgery, Seoul National University Hospital, Seoul 03080, Korea; 10Division of Gynecologic Oncology, Department of Obstetrics and Gynecology, Ajou University School of Medicine, Suwon 16499, Korea; 11Agaon Fertility Clinic, Seoul 08391, Korea; 12Department of Obstetrics and Gynecology, Seoul National University College of Medicine, Seoul 03080, Korea

**Keywords:** peritoneal metastasis, rotational intraperitoneal pressurized aerosol chemotherapy, nozzle position, posture, distribution, penetration

## Abstract

Even though rotational intraperitoneal pressurized aerosol chemotherapy (RIPAC) has been developed to improve the distribution and penetration depth of anti-cancer agents by pressurized intraperitoneal aerosol chemotherapy (PIPAC), the optimal nozzle position and patient’s posture have not been investigated. Thus, we used nine pigs weighing 50–60 kg, and sprayed 150 mL of 1% methylene blue as an aerosol through the nozzle, DreamPen^®^ (Dreampac Corp., Wonju, Republic of Korea), with a flow rate of 0.6 ml/min under a pressure of 140 to 150 psi for RIPAC in six and three pigs with supine and Trendelenburg positions, respectively. When we evaluated its distribution and penetration depth, even distribution among 13 regions of the abdomen was observed in three pigs with Trendelenburg position regardless of the depth of the nozzle. Regarding penetration depth, the numbers of regions with maximal penetration depth were high in the 2 cm depth of the nozzle with supine position (*n* = 5) and the 4 cm depth with Trendelenburg position (*n* = 3). Conclusively, even distribution and maximal penetration of anti-cancer agents can be expected during RIPAC in the medium depth (4 cm) between the nozzle inlet and the visceral peritoneum located on the opposite side of it and the Trendelenburg position.

## 1. Introduction

Peritoneal metastasis (PM) from solid tumors is difficult to treat because PM is shown as an advanced or recurrent disease with resistance to various types of anti-cancer agents [1,2]. The prognosis of PM is very poor, and if left untreated, most patients die despite active treatment within several months [3,4]. Especially, research results showing that systemic chemotherapy is no longer effective due to the development of various mechanisms to evade cytotoxicity limit the treatment option of PM, whereas increased toxicities to systemic chemotherapy play an important role in the transition from treatment to palliative care [5,6].

In contrast, intraperitoneal chemotherapy has recently attracted attention in patients with PM because it shows its effect by diffusion to the peritoneum with less toxicity than systemic chemotherapy [7]. Even though hyperthermic intraperitoneal chemotherapy (HIPEC) has emerged for treating PM, there are some debates about its effect on treating PM because HIPEC should be combined with cytoreductive surgery to remove almost all of the tumors, and its effect may vary depending on the tumor origin and histologic type [8,9,10].

On the other hand, pressurized intraperitoneal aerosol chemotherapy (PIPAC) is considered as an alternative to compensate for these shortcomings of HIPEC [11]. PIPAC as a laparoscopic intraperitoneal chemotherapy spray 10% dose of anti-cancer agents used in systemic chemotherapy in the form of aerosol made by a high-pressure injector, and thereby tissue concentration of anti-cancer agents reaches about 200 times the blood concentration with minimal toxicities such as postoperative pain, suggesting the potential to overcome drug resistance [12]. However, PIPAC has some disadvantages including uneven distribution and limited penetration of anti-cancer agents because most anti-cancer agents sprayed as an aerosol are mainly delivered to the opposite side of the nozzle [13,14].

Thus, the Korean Rotational Intraperitoneal Pressurized Aerosol Chemotherapy (KoRIA) trial group developed rotational intraperitoneal pressurized aerosol chemotherapy (RIPAC) to enhance drug delivery by adding conical pendulum motion of the nozzle, DreamPen^®^ (Dreampac Corp., Wonju, Republic of Korea) [15,16]. As a result, RIPAC improved tissue concentration and penetration of anti-cancer agents in ex vivo and in vivo models [16,17], and the position of DreamPen^®^ (Dreampac Corp., Wonju, Republic of Korea) was advantageous for drug delivery when it was located halfway between the nozzle inlet and the bottom in an ex vivo model [18].

Nevertheless, the optimal nozzle position and patient’s posture for RIPAC have not been investigated. Thus, we designed this experimental study to evaluate them in an in vivo model before the actual clinical trial to improve drug delivery during RIPAC was conducted.

## 2. Materials and Methods

We used a total of nine pigs weighing 50 to 60 kg with an abdominal cavity similar to women for this research. Among them, six and three received RIPAC in the supine and Trendelenburg positions, respectively. For RIPAC, we made capnoperitoneum by insufflating CO_2_ via a Veress needle to each pig, and then inserted two 12 mm bladeless trocars (Eagleport^®^; Dalim Medical Corp., Seoul, Republic of Korea) to use passages for inserting DreamPen^®^ (Dreampac Corp., Wonju, Republic of Korea) and laparoscopic devices (Stryker Korea Co., Ltd., Seoul, Republic of Korea) 10 cm above and below the median point between the low margin of the sternum and the joint connecting the femur and ilium.

For evaluating drug distribution and penetration by RIPAC, we prepared 150 mL of 1% methylene blue, which was sprayed as an aerosol with a median diameter of 30 μm via DreamPen^®^ (Dreampac Corp., Wonju, Republic of Korea) with a flow rate of 0.6 ml/min under a pressure of 140 to 150 psi made by Illumena^®^ Néo (Geurbet Korea Co. Ltd., Seoul, Republic of Korea). Moreover, DreamPen^®^ (Dreampac Corp., Wonju, Republic of Korea) was placed at a depth of 2, 4, and 7 cm from the visceral peritoneum located on the opposite side of it. After completion of RIPAC, a capnoperitoneum of 12 mmHg was maintained for 30 min, and the pigs were euthanized.

We used the modified Peritoneal Cancer Index (PCI) including the central, right upper quadrant (RUQ), epigastrium, left upper quadrant (LUQ), left flank (LF), left lower quadrant (LLQ), pelvic, right lower quadrant (RLQ), right flank (RF), ileum, colon, stomach, and jejunum regions for investigating drug delivery to the parietal peritoneum of pigs [16,19]. For evaluating the distribution of 1% methylene blue, we scored the intensity from 0 to 3 points (no stain, weak, moderate and strong) and percentage from 0 to 4 points (0%, <25%, 25–50%, 50–75%, >75%). The final score of the distribution was determined by multiplying these two scores. The two authors (D.W.H. and D.K.) evaluated them, and any discrepancies were addressed by a joint re-evaluation with the third author (H.S.K.). To investigate its penetration depth, we used a caliper capable of measuring precision down to 1/100 of a millimeter. Then, three tissues were obtained from each region to measure the penetration depth three times. The two authors (E.J.L. (Eun Joo Lee) and J.Y.C.) investigated them, and inconsistencies were resolved through joint re-evaluation with the third author (S.J.P.; Figure 1).

For statistical analyses, we used SPSS version 25 software (IBM Corp., Armonk, NY, USA, RRID:SCR_002865), and *p* ≤ 0.05 was considered significant because of the non-parametric tests.

## 3. Results

In terms of the distribution, scores among the 2, 4, and 7 cm depth of the nozzle were not different in all regions when postures were not distinguished (Table 1).

When we evaluated the distribution among 13 regions, scores were not different in the 2, 4, and 7 cm depth of the nozzle. In six pigs with the supine position, scores were not different in 2 and 7 cm depth, whereas uneven distribution was shown in the 4 cm depth of the nozzle. On the other hand, the distribution was not different among 13 regions in 2, 4, and 7 cm depth of the nozzle for three pigs with the Trendelenburg position (Figure 2).

When we compared the penetration depth among the 2, 4, and 7 cm depths of the nozzle in 13 regions, it was different in the LUQ, LF, and jejunum regions for all pigs. In six pigs with the supine position, the penetration depth was different in the central, RUQ, LUQ, LF, and jejunum regions, whereas it was different in the central, epigastrium, colon, and jejunum regions of three pigs with the Trendelenburg position (Table 2).

To summarize, the numbers of regions with maximal penetration depth appeared to be high in the 2 cm depth of the nozzle with the supine position (*n* = 5) and 4 cm depth with the Trendelenburg position of the nozzle (*n* = 3; Figure 3).

## 4. Discussion

RIPAC was developed to improve the uneven distribution of anti-cancer agents with PIPAC spray aerosol over a wider range by adding the rotational motion of the nozzle. As a result, RIPAC overcame the limitation of distributing most aerosol on the opposite side of the nozzle during PIPAC, and efficiently delivered drugs to various regions within the abdomen in preclinical studies.

Research on the distribution and penetration depth of anti-cancer agents by PIPAC reported the highest values on the opposite side of the nozzle, showing maximal penetration depth ranging from 300 to 400 μm in the opposite side of the nozzle and minimal penetration depth less than 100 μm [13,14,20]. Even though the depth of the nozzle and the dosage of anti-cancer agents could improve the penetration depth, the maintenance pressure in the cavity failed to improve it. In particular, the three conditions were not factors for improving the distribution of anti-cancer agents in PIPAC [13].

Nevertheless, RIPAC enhanced drug delivery, compared with PIPAC in an ex vivo model [18]. We thought that reduced turbulent flow by a lower velocity of aerosol and larger injection outlet size of DreamPen^®^ (Dreampac Corp., Wonju, Republic of Korea) might lead to the wider movement of the aerosol through an increase in deflection. In this in vivo research, we also found even distribution after RIPAC, especially, in the Trendelenburg position, showing no difference in the intensity and percentage in 13 regions. It means that Trendelenburg position can be considered the optimal posture for RIPAC.

In addition, the numbers of regions with maximal penetration depth were high in the 2 cm depth of the nozzle with the supine position (*n* = 5) and 4 cm depth with the Trendelenburg position of the nozzle (*n* = 3). Even though the 2 cm depth of the nozzle in the supine position looks to be more beneficial, especially in central, RUQ, LUQ, LF, and Jejunum, the distribution scores were lower in the epigastrium, LUQ, stomach, and jejunum regions. This suggests the potential that a relatively narrow distribution of anti-cancer agents during RIPAC cannot control PM despite their high penetration depth. Therefore, we thought that the 4 cm depth of the nozzle in the Trendelenburg position might be ideal for better distribution and penetration depth of anti-cancer agents based on the results of this research.

Nevertheless, this study has some limitations as follows: First, tumor location and intra-abdominal adhesion can change patterns of the distribution and penetration depth of anti-cancer agents during RIPAC. Second, delayed uptake by tumors and lymphatic absorption can also lead to uneven distribution and accumulation of aerosol [18]. Third, we used only 1% methylene blue and measured its distribution and penetration depth macroscopically in pigs. Thus, further studies where anti-cancer agents including doxorubicin, cisplatin, and oxaliplatin are used during RIPAC should be conducted to support these results. Fourth, the optimal position and patient’s posture for RIPAC should be compared with those for PIPAC. However, we could not perform the traditional PIPAC as a control because of a lack of the device in our country. Fifth, further evidence to support these results is needed in a clinical setting.

## 5. Conclusions

In conclusion, even distribution and maximal penetration of anti-cancer agents can be expected during RIPAC in the medium depth (4 cm) between the nozzle inlet and the visceral peritoneum located on the opposite side of it and the Trendelenburg position.

## 6. Patents

S.J.P. and H.S.K. have a relevant patent (No. 1020210042898, Republic of Korea; PCT/KR2021/006829).

## Figures and Tables

**Figure 1 jpm-12-01799-f001:**
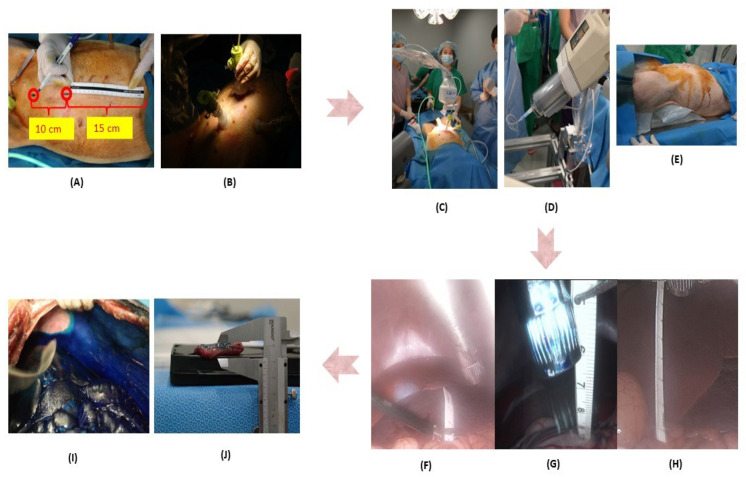
The experimental process for evaluating the optimal nozzle position and patient posture for rotational intraperitoneal pressurized aerosol chemotherapy (RIPAC): (**A**) insertion positions of two 12 mm bladeless trocars (the first position was 15 cm below the xyphoid process, and the second position was located 10 cm further below); (**B**) the view after insertion of two 12 mm bladeless trocars; (**C**) the view after installation of RIPAC system; (**D**) a high-pressure injector for spraying 150 ml of 1% methylene blue as aerosol; (**E**) Trendelenburg position of pigs; Dreampen^®^ (Dreampac Corp., Wonju, Republic of Korea) placed at a depth of 2 cm (**F**), 4 cm (**G**), and 7 cm (**H**) from the visceral peritoneum; Evaluation of the distribution (**I**) and penetration depth (**J**) of 1% methylene blue after RIPAC.

**Figure 2 jpm-12-01799-f002:**
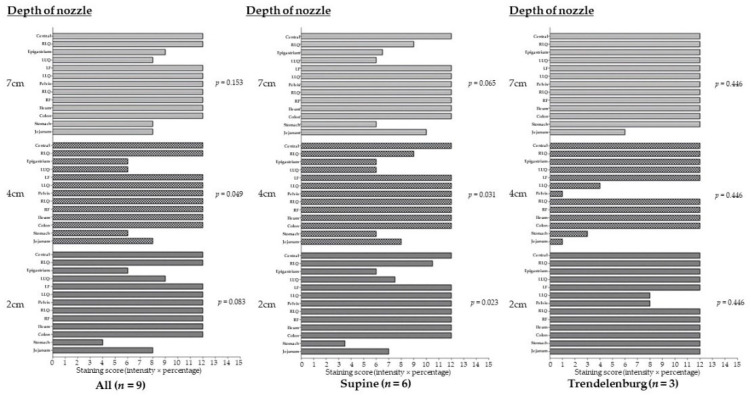
Comparison of the distribution of 1% methylene blue among 13 regions of pigs’ abdomen in the 2, 4, and 7 cm depth of the nozzle from the visceral peritoneum.

**Figure 3 jpm-12-01799-f003:**
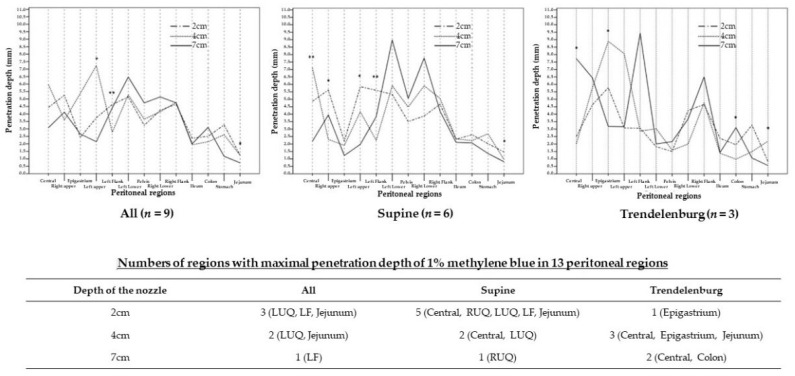
Number of regions with the maximal penetration depth of 1% methylene blue in 13 regions of pigs’ abdomen based on the depth of the nozzle.

**Table 1 jpm-12-01799-t001:** Comparison of the distribution of 1% methylene blue among 2, 4, and 7 cm depth of the nozzle in 13 regions of pigs’ abdomen.

Regions	The Depth of the Nozzle			*p* Value
	2 cm	4 cm	7 cm	
Central	12 (12, 12)	12 (12, 12)	12 (12, 12)	1.000
RUQ	12 (9, 12)	12 (6, 12)	12 (6, 12)	0.953
Epigastrium	6 (6, 12)	6 (6, 12)	9 (4, 12)	1.000
LUQ	9 (6, 12)	6 (6, 12)	8 (4, 12)	0.867
LF	12 (12, 12)	12 (12, 12)	12 (12, 12)	1.000
LLQ	12 (8, 12)	12 (4, 12)	12 (12, 12)	0.558
Pelvis	12 (8, 12)	12 (1, 12)	12 (12, 12)	0.558
RLQ	12 (12, 12)	12 (12, 12)	12 (12, 12)	1.000
RF	12 (12, 12)	12 (12, 12)	12 (12, 12)	1.000
Ileum	12 (12, 12)	12 (12, 12)	12 (12, 12)	1.000
Colon	12 (12, 12)	12 (12, 12)	12 (12, 12)	1.000
Stomach	4 (3, 12)	6 (3, 6)	8 (4, 12)	0.569
Jejunum	8 (6, 12)	8 (1, 8)	8 (6, 12)	0.717

Abbreviations: LF, left flank; LLQ, left lower quadrant; LUQ, left upper quadrant; RF, right flank; RLQ, right lower quadrant; RF, right flank; RUQ, right upper quadrant.

**Table 2 jpm-12-01799-t002:** Comparison of the penetration depth of 1% methylene blue among 2, 4, and 7 cm depth of the nozzle in 13 regions of the pigs’ abdomen.

Regions	The Depth of the Nozzle			*p* Value
	2 cm	4 cm	7 cm	
**All**				
Central	4.44 (2.00, 6.67)	5.96 (1.75, 9.03)	3.08 (1.28, 9.09)	0.591
RUQ	5.25 (3.08, 7.14)	3.59 (0.63, 6.00)	4.12 (3.55, 6.45)	0.143
Epigastrium	2.43 (1.29, 6.67)	5.38 (0.88, 11.84)	2.6 (0.65, 4.09)	0.270
LUQ	3.75 ^a^ (2.31, 11.39)	7.23 ^a,c^ (0.33, 10.28)	2.14 ^c^ (1.33, 7.11)	0.042
LF	4.63 ^b^ (1.94, 8.37)	2.78 (1.43, 3.61)	4.41 ^b^ (1.75, 9.44)	0.006
LLQ	5.14 (0.91, 6.38)	5.28 (1.21, 7.73)	6.47 (1.43, 12.00)	0.426
Pelvis	3.24 (1.00, 6.73)	3.65 (1.36, 7.11)	4.74 (1.89, 6.84)	0.365
RLQ	4.25 (2.40, 6.36)	4.15 (1.60, 10.77)	5.14 (2.63, 10.53)	0.454
RF	4.65 (1.86, 6.00)	4.77 (1.36, 5.95)	4.74 (2.25, 7.25)	0.686
Ileum	2.38 (1.56, 3.30)	1.94 (1.03, 2.89)	2.00 (1.43, 2.31)	0.129
Colon	2.50 (1.71, 4.08)	2.14 (0.59, 2.82)	3.10 (1.09, 4.85)	0.102
Stomach	3.27 (0, 5.74)	2.61 (1.06, 2.86)	1.17 (0.64, 5.12)	0.313
Jejunum	1.25 ^a^ (0.83, 2.45)	1.25 ^a,c^ (0.63, 2.19)	0.73 ^c^ (0.56, 1.32)	0.048
**Supine position**				
Central	4.87 ^a^ (4.44, 6.67)	7.12 ^a^ (3.94, 9.03)	2.18 (1.28, 3.08)	0.002
RUQ	5.62 ^b^ (3.59, 7.14)	2.31 (0.63, 5.22)	3.95 ^b^ (3.68, 6.29)	0.019
Epigastrium	2.19 (1.29, 2.94)	1.91 (0.88, 8.04)	1.23 (0.65, 2.93)	0.420
LUQ	5.84 ^a^ (3.33, 11.39)	4.17 ^a,c^ (0.33, 7.44)	1.98 ^c^ (1.33, 2.89)	0.024
LF	5.61 (4.15, 8.37)	2.24 ^c^ (1.43, 3.61)	3.84 ^c^ (1.75, 5.14)	0.005
LLQ	5.33 (5.00. 6.38)	5.91 (4.09. 7.73)	8.97 (4.71, 12.00)	0.090
Pelvis	3.51 (3.16, 6.73)	4.48 (2.69, 7.11)	5.06 (2.37, 6.84)	0.587
RLQ	3.88 (2.40, 6.36)	5.91 (2.93, 10.77)	7.77 (4.05, 10.53)	0.115
RF	4.67 (3.81, 6.00)	5.06 (1.88, 5.95)	4.19 (2.25, 7.25)	0.630
Ileum	2.32 (1.56, 3.39)	2.34 (1.88, 2.89)	2.11 (1.78, 2.31)	0.584
Colon	2.63 (2.36, 4.08)	2.24 (1.92, 2.82)	2.07 (1.09, 4.85)	0.413
Stomach	2.02 (0, 5.74)	2.69 (1.90, 2.86)	1.37 (0.67, 5.12)	0.674
Jejunum	1.47 (1.11, 2.45)	0.96 ^c^ (0.63, 1.54)	0.81 ^c^ (0.67, 1.32)	0.028
**Trendelenburg position**				
Central	2.50 ^a,b^ (2.00, 3.00)	2.00 ^a,c^ (1.75, 2.25)	7.73 ^b,c^ (6.59, 9.09)	0.044
Right upper	4.62 (3.08, 5.64)	5.75 (4.00, 6.00)	6.45 (3.55, 6.45)	0.390
Epigastrium	5.78 ^a^ (5.33, 6.67)	8.89 ^a^ (7.22, 11.94)	3.18 (3.18, 4.09)	0.027
Left upper	3.08 (2.31, 5.13)	8.06 (7.78, 10.28)	3.16 (2.63, 7.11)	0.061
Left flank	3.06 (1.94, 4.44)	2.86 (1.71, 3.14)	9.44 (7.78, 9.44)	0.059
Left lower	1.82 (0.91, 2.27)	3.03 (1.21, 3.33)	2.00 (1.43, 2.29)	0.491
Pelvis	1.50 (1.00, 1.50)	1.59 (1.35, 3.41)	2.16 (1.89, 6.49)	0.111
Right lower	4.25 (3.00, 5.50)	2.00 (1.60, 3.20)	3.68 (2.63, 3.68)	0.146
Right flank	4.65 (1.86, 4.88)	4.77 (1.36, 5.23)	6.50 (6.00, 7.00)	0.066
Ileum	2.38 (2.14, 2.86)	1.38 (1.03, 1.72)	1.43 (1.43, 1.90)	0.050
Colon	1.95 (1.71, 2.44)	0.98 (0.59, 0.98)	3.10 (3.10, 3.79)	0.026
Stomach	3.27 (2.65, 3.67)	1.49 (1.06, 2.77)	1.06 (0.64, 1.28)	0.067
Jejunum	0.83 ^b^ (0.83, 1.25)	2.19 (1.88, 2.19)	0.56 ^b^ (0.56, 1.11)	0.047

Abbreviations: LF, left flank; LLQ, left lower quadrant; LUQ, left upper quadrant; RF, right flank; RLQ, right lower quadrant; RF, right flank; RUQ, right upper quadrant. All values were median and range (mm), and values labeled with the same characters are not significantly different.

## Data Availability

The data presented in this research are available on request from the corresponding author.

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
