# Peer review of "Optimal Nozzle Position and Patient’s Posture to Enhance Drug Delivery into the Peritoneum during Rotational Intraperitoneal Pressurized Aerosol Chemotherapy in a Swine Model"

_jpm, 2022, doi:10.3390/jpm12111799_

Round 1

Reviewer 1 Report

This in vivo experiment is interesting and could have real clinical implications in PIPAC and RIPAC treatment implimentation. However, further evidence is needed in clinical setting

Author Response

Q1. This in vivo experiment is interesting and could have real clinical implications in PIPAC and RIPAC treatment implimentation. However, further evidence is needed in clinical setting

A1. Thank you for your good comments. Your point is very important because the effect of RIPAC compared with PIPAC may be meaningful in clinical setting. So, we mentioned relevant contents in Discussion as follows: Line 216-217 “Fifth, further evidence to support these results is needed in clinical setting.”

Reviewer 2 Report

1. The authors have made a good trial of the revised technique of PIPAC in the administration of chemotherapy drugs. However, the description of the process is ambiguous. Please add a graph or a link connecting to a video for the authors to grasp the key points of the procedure.

2. I suggest the authors use the traditional PIPAC technique under the same circumstance as the control group to acquire the result for comparison with their modified technique.

3. The similarity rate is 17% after the check with the Turnitin system, which is acceptable.

Author Response

Q1. The authors have made a good trial of the revised technique of PIPAC in the administration of chemotherapy drugs. However, the description of the process is ambiguous. Please add a graph or a link connecting to a video for the authors to grasp the key points of the procedure.

A1. Thank you for your good recommendation! As you recommended, we added the experimental process as figure 1.

Q2. I suggest the authors use the traditional PIPAC technique under the same circumstance as the control group to acquire the result for comparison with their modified technique.

A2. Good recommendation! But, we could not get the traditional PIPAC system, which has different devices from our RIPAC system. In the current status, it is impossible to compare the optimal position and patent posture between the treatments. So, we mentioned this limitation in the Discussion as follows: Line 214-216 “Fourth, the optimal position and patient posture for RIPAC should be compared with those for PIPAC. However, we could not perform the traditional PIPAC as a control because of a lack of the device in our country.”

Q3. The similarity rate is 17% after the check with the Turnitin system, which is acceptable.

A3. We deeply appreciate your valuable review for checking the similarity rate in this manuscript.

Round 2

Reviewer 2 Report

The authors have revised the manuscript according to the suggestions.